# A substrate-driven allosteric switch that enhances PDI catalytic activity

Roelof H. Bekendam[1], Pavan K. Bendapudi[1], Lin Lin[1], Partha P. Nag[2,3], Jun Pu[2], Daniel R. Kennedy[4], Alexandra Feldenzer[4], Joyce Chiu[5,6], Kristina M. Cook[5], Bruce Furie[1], Mingdong Huang[1], Philip J. Hogg[5,6] & Robert Flaumenhaft[1]

Protein disulfide isomerase (PDI) is an oxidoreductase essential for folding proteins in the endoplasmic reticulum. The domain structure of PDI is **a**–**b**–**b**′–**x**–**a**′, wherein the thioredoxin-like **a** and **a**′ domains mediate disulfide bond shuffling and **b** and **b**′ domains are substrate binding. The **b**′ and **a**′ domains are connected via the x-linker, a 19-amino-acid flexible peptide. Here we identify a class of compounds, termed bepristats, that target the substrate-binding pocket of **b**′. Bepristats reversibly block substrate binding and inhibit platelet aggregation and thrombus formation *in vivo*. Ligation of the substrate-binding pocket by bepristats paradoxically enhances catalytic activity of **a** and **a**′ by displacing the x-linker, which acts as an allosteric switch to augment reductase activity in the catalytic domains. This substrate-driven allosteric switch is also activated by peptides and proteins and is present in other thiol isomerases. Our results demonstrate a mechanism whereby binding of a substrate to thiol isomerases enhances catalytic activity of remote domains.

[1] Division of Hemostasis and Thrombosis, Department of Medicine, Beth Israel Deaconess Medical Center, Harvard Medical School, Boston, Massachusetts 02115, USA. [2] Department of Pharmaceutical and Administrative Sciences, The Broad Institute Probe Development Center, Cambridge, Massachusetts 02142, USA. [3] Center for the Science of Therapeutics, Broad Institute, Cambridge, Massachusetts 02142, USA. [4] College of Pharmacy, Western New England University, Springfield, Massachusetts 01119, USA. [5] The Centenary Institute, Newtown, Sydney, New South Wales 2050, Australia. [6] National Health and Medical Research Council Clinical Trials Centre, University of Sydney, Sydney, New South Wales 2050, Australia. Correspondence and requests for materials should be addressed to R.F. (email: rflaumen@bidmc.harvard.edu).

Protein disulfide isomerase (PDI) is the founding member of a large family of thiol isomerases responsible for catalysing the folding of nascent proteins in the endoplasmic reticulum (ER). Although highly concentrated in the ER, a small percentage of PDI escapes ER retention and traffics to secretory granules and plasma membrane, where it participates in modification of surface proteins. The domain structure of PDI is **a**–**b**–**b′**–**x**–**a′** (refs 1,2). Disulfide shuffling required for protein folding is accomplished by active site cysteines within a CGHC motif in the thioredoxin-like **a** and **a′** domains. Substrate binding is accomplished by domains **b** and **b′**, the latter of which contains a deep hydrophobic binding pocket[1,2]. The x-linker is a flexible 19–amino-acid peptide that can adopt at least two conformations. One is a 'capped' conformation in which the x-linker covers the hydrophobic pocket[3]. PDI can also assume an 'uncapped' conformation in which the x-linker is displaced from the hydrophobic binding site. Although distinct functions have been described for the different domains of PDI, there is cooperativity among them[1,4,5]. The presence of the **b** and **b′** domains in PDI augments reductase activity of the **a** and **a′** domains[5]. Conversely, the **a** and **a′** domains are important for binding larger substrates.

In addition to its physiological role in protein folding, PDI has been implicated in a wide variety of pathophysiological processes. PDI expression is upregulated in several cancers[6], and PDI expression levels correlate with clinical outcomes[7,8]. Silencing or inhibition of PDI in animal models of tumour progression suppresses tumour growth and extends survival[9,10]. PDI has also been shown to participate in neurodegenerative processes[11], and blocking PDI is protective in a cell-based model of Huntington's disease (refs 12,13). Several pathogens subvert extracellular PDI activity to achieve cellular invasion[14–17]. For example, PDI mediates cleavage of disulfide bonds in glycoprotein 120 that are required for HIV-1 entry[18,19], and its inhibition interferes with the ability of HIV-1 to infect T cells[20]. Extracellular PDI also serves a critical role in thrombus formation, the underlying pathology in myocardial infarction, stroke, peripheral artery disease and deep vein thrombosis. Inhibition of extracellular PDI blocks injury-induced formation of thrombi in multiple animal models of thrombus formation[21–25]. Platelet-specific knockdown of PDI inhibits thrombus formation, demonstrating a role for platelet-derived PDI in thrombus formation[26]. Two other members of the thiol isomerase family, ERp5 and ERp57 (which, together with PDI, are termed 'vascular thiol isomerases') are important for thrombosis[27–31].

Several novel PDI inhibitors have been identified over the past decade. The majority of these antagonists act at the catalytic cysteine within the CxxC motif, blocking all catalytic activity of PDI, and most of these antagonists act irreversibly[6,32]. As a class, compounds that interact with the catalytic cysteines of PDI are not selective among thiol isomerases. The substantial homology in the thioredoxin folds of the catalytic domains of thiol isomerases[33] complicates efforts to develop compounds that are selective among this large enzyme family. As substrate-binding domains are less homologous and the function of PDI relies on cooperative activity of distinct domains, alternative strategies for blocking PDI that do not involve inhibition of the catalytic cysteines could be effective.

With support from the Molecular Libraries Program, we screened over 348,505 compounds from the Molecule Libraries Small Molecule Repository to identify inhibitors of PDI reductase activity[34,35]. We characterized two compounds, termed bepristats, based on their unusual activity at the **b′** domain. We find that binding of bepristats to the hydrophobic pocket in the **b′** domain of PDI interferes with insulin binding and displaces the x-linker that covers this pocket. Displacement of the x-linker induces a conformation change, modifying the redox state of the catalytic motif and enhancing PDI catalytic activity as measured by di-eosin-GSSG cleavage. Peptides known to bind PDI, such as mastoparan and somatostatin and protein substrates (for example, cathespin G) also triggered this substrate-driven allosteric switch. These results support a general mechanism wherein ligation of the substrate-binding domain of PDI enhances its catalytic activity.

## Results

**Characterization of novel PDI antagonists.** High-throughput screening of 348,505 compounds identified two scaffolds that inhibited PDI reductase activity in the insulin turbidimetric assay. The first scaffold, represented by CID:23723882/ML359 (refs 34,35) and a synthetically prepared analogue, termed bepristat 1a (CID: 70701262), blocked PDI activity with a half-maximal inhibitory concentration ($IC_{50}$) of $\sim 0.7\,\mu M$ (Fig. 1a). A second scaffold characterized by CID:1043221 (ref. 29) and a synthetically prepared analogue bepristat 2a (CID: 71627360) was found to have an $IC_{50}$ of $\sim 1.2\,\mu M$ (Fig. 1b). Yet despite potent PDI inhibitory activity in the insulin turbidimetric assay, neither bepristat 1a nor bepristat 2a inhibited in a di-eosin-GSSG-based assay that measures reductase activity at the catalytic cysteines[36] (Fig. 1a,b). In fact, rather than inhibiting reductase activity at the catalytic cysteines, bepristat 1a and bepristat 2a both enhanced cleavage of the di-eosin-GSSG probe by PDI (Fig. 1a,b). Neither bepristat 1a nor bepristat 2a affected the fluorescence of di-eosin-GSSG in the absence of PDI, nor did these compounds affect di-eosin-GSSG fluorescence in the presence of a catalytically inactive PDI (Fig. 1c and Supplementary Fig. 1). Michaelis–Menten analysis performed in the absence of bepristats showed an apparent $K_M$ of 4,168 nM and an apparent $k_{cat}$ of $1,135\,min^{-1}$ for cleavage of di-eosin-GSSG by PDI. In the presence of bepristat 1a, the apparent $K_M$ was decreased to 1,814 nM and apparent $k_{cat}$ was unaffected, while incubation with bepristat 2a decreased the apparent $K_M$ to 1,949 nM and increased the apparent $k_{cat}$ to $1,995\,min^{-1}$ (Fig. 1d). Bepristats demonstrated more potent inhibitory activity in the insulin turbidimetric assay compared with quercetin-3-rutinoside (rutin), a glycosylated flavonoid quercetin shown to block PDI activity and inhibit thrombus formation *in vivo*[22,37]; PACMA-31, an irreversible inhibitor of PDI that binds the catalytic cysteines and impairs tumour growth in a murine model of ovarian cancer[9]; and bacitracin, a dodecapeptide that has been used for decades as the standard PDI inhibitor[38] (Fig. 1e). Yet these other inhibitors all blocked PDI-mediated cleavage of di-eosin-GSSG when used at concentrations required to inhibit activity in the insulin turbidimetric assay (Fig. 1e). These results indicate that bepristats block PDI reductase activity via a mechanism that differs from that of previously described PDI inhibitors.

**Selectivity and reversibility of bepristats.** Developing selective, reversible inhibitors of PDI has been problematic owing to homology among catalytic thioredoxin domains and the ability of thiol-reactive antagonists to interact with multiple targets. We compared bepristats and thiol-reactive PDI inhibitors in the insulin turbidimetric assay to assess their ability to inhibit the insulin reductase activities of ERp5, ERp57 and thioredoxin. We found that bepristats were selective among vascular thiol isomerases, even at concentrations 10-fold higher than their $IC_{50}$s (Fig. 2a and Supplementary Fig. 1). In contrast, PACMA-31 demonstrated inhibitory activity against both ERp5 and thioredoxin when used at 10-fold their $IC_{50}$ (Fig. 2b). Bacitracin showed activity against all thiol isomerases tested. Both PACMA-31 and bacitracin interfered with the reaction of

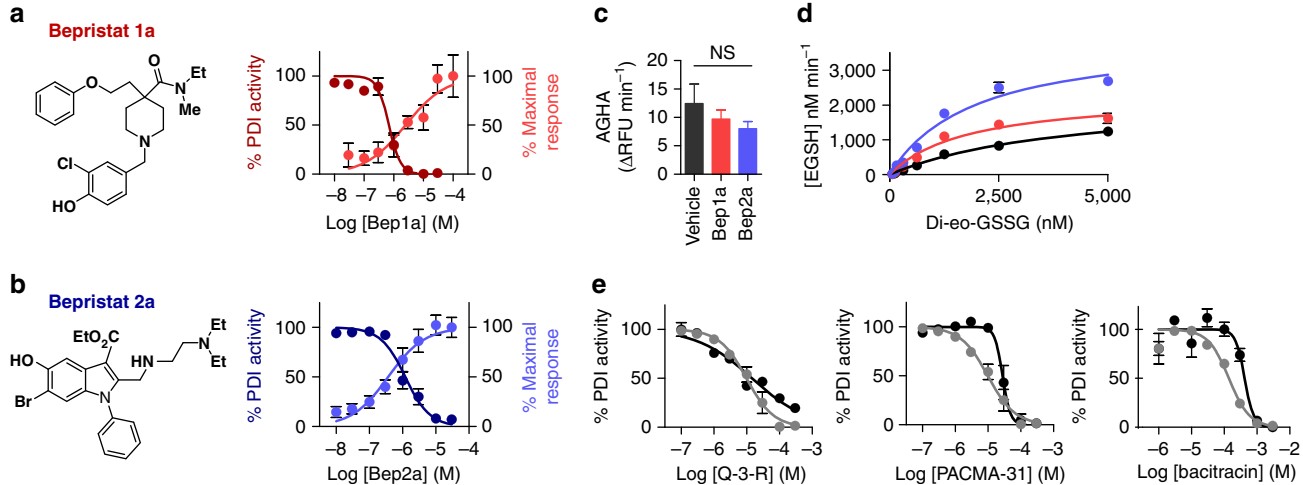

**Figure 1 | Novel PDI antagonists paradoxically augment catalytic activity.** The effect of (**a**) bepristat 1a (red) and (**b**) bepristat 2a (blue) on PDI activity as measured in the insulin turbidimetric assay (dark shades) or the di-eosin-GSSG assay (light shades). Curves represent absorbance values taken from the 25 min time point after insulin aggregation started in the vehicle only sample. (**c**) Bepristat 1a (Bep1a) and bepristat 2a (Bep2a) fail to augment activity of a PDI mutant in which the CGHC motif is mutated to AGHA. Data are represented as cleavage of di-eosin-GSSG probe in RFU min$^{-1}$. One-way analysis of variance with Dunnett's post test: NS, non-significant. (**d**) The effect of bepristat 1a (red) and bepristat 2a (blue) on $K_M$ and $k_{cat}$ was evaluated in the di-eosin-GSSG assay using increasing concentrations of di-eosin-GSSG. Michaelis–Menten kinetics were calculated using Graphpad Prism 5.0 and compared with vehicle only (black). (**e**) Effect of quercetin-3-rutinoside (Q-3-R), PACMA-31 or bacitracin on PDI activity as measured by the insulin turbidimetric assay (black) or the di-eosin-GSSG assay (grey). Values for the insulin turbidimetric assay represent per cent of PDI activity compared with a control exposed to vehicle alone, mean ± s.e.m. ($n = 3–6$). Values for the di-eosin-GSSG assay represent the rate of di-eosin-GSSG fluorescence generation measured for 20 min ± s.e.m. ($n = 3$). RFU, relative fluorescence units.

maleimide-polyethylene glycol-2-biotin (MPB) with the catalytic cysteines of ERp5, ERp57 and thioredoxin (Fig. 2c and Supplementary Figs 2 and 3), demonstrating lack of selectivity by virtue of non-selective interactions with catalytic cysteines on thiol isomerases. When the assay is performed in the presence of *N*-ethylmaleimide, an irreversible free thiol blocker, no MPB binding to PDI is observed, indicating absence of non-specific MPB labelling of PDI. None of the PDI inhibitors interfered with the MPB reaction with bovine serum albumin (BSA), demonstrating that the inhibitors are not non-selectively reacting with MPB or with any free cysteine.

Inhibition of PDI by either blocking antibodies or small-molecule inhibitors interferes with agonist-induced platelet activation[21,22]. Platelet-specific knockdown of PDI also impairs platelet activation[26]. We determined whether blocking PDI using bepristats inhibits platelet activation. Platelets were incubated with either bepristat 1b (a bepristat 1 analogue with enhanced esterase resistance necessary to overcome intrinsic platelet esterase activity), bepristat 2a or PACMA-31, and the response to the PAR1 peptide agonist SFLLRN was evaluated by light transmission aggregometry. Bepristat 1b, bepristat 2a and PACMA-31 all inhibited platelet aggregation (Fig. 2d). We next evaluated whether or not the bepristats functioned as global inhibitors of platelet activation. Neither bepristat significantly altered expression of CD62P on stimulated platelets, indicating that platelet activation *per se* is not blocked by the bepristats in this assay (Supplementary Fig. 4; Supplementary Methods). Rather, bepristats appear to block aggregation by interfering with functions downstream of platelet activation. To evaluate reversibility of inhibition using the platelet aggregation assay, platelets were incubated with PDI antagonists for 30 min, washed and then stimulated with SFLLRN. Inhibition of platelet aggregation by bepristat 1b and bepristat 2a was restored following washing. In contrast, platelet aggregation by PACMA-31 was irreversibly inhibited under these conditions (Fig. 2). To confirm that bepristats are reversible inhibitors of PDI, we

evaluated reversibility in the insulin turbidimetric assay. These studies demonstrated that the inhibitory effect of bepristats was readily reversed by dilution to a subinhibitory concentration, while that of PACMA-31 was largely preserved (Supplementary Fig. 5).

**Bepristats inhibit thrombus formation.** Inhibition of PDI using anti-PDI antibodies or by small molecules such as bacitracin or quercetin-3-rutinoside inhibits thrombus formation *in vivo*[21–25]. Similarly, platelet-selective deletion of PDI interferes with thrombus formation in mouse arterioles following vascular injury[26]. We therefore evaluated the effect of bepristats on thrombus formation in cremaster arterioles following laser-induced vascular injury (Fig. 3a and Supplementary Movie 1). Infusion of 15 mg kg$^{-1}$ of bepristat 1a resulted in a 79.7% reduction in platelet accumulation at sites of vascular injury compared with mice infused with vehicle alone ($P = 0.02$; Fig. 3b and Supplementary Movie 2). Bepristat 2a infusion inhibited platelet accumulation by 85.1% at sites of laser-induced injury compared with vehicle controls ($P = 0.02$; Fig. 3c and Supplementary Movie 3). These results demonstrate that bepristats are tolerated *in vivo* and potently inhibit thrombus formation.

**Bepristats associate with the b' domain.** To determine the mechanism by which bepristat 1a and bepristat 2a modulate PDI activity, we tested the compounds against PDI fragments containing the **a** or **a'** domains using the insulin turbidimetric assay. These fragments included the **a** domain, **a'** domain, **ab** domains, **abb'** domains and **b'xa'** domains (Fig. 4a and Supplementary Fig. 6). Although the isolated domains had diminished insulin reductase activity compared with full-length PDI, their activity could be quantified and the effects of antagonists on their activity tested. Neither bepristat 1a nor bepristat 2a had activity against the isolated **a**, **a'** or **ab** domains

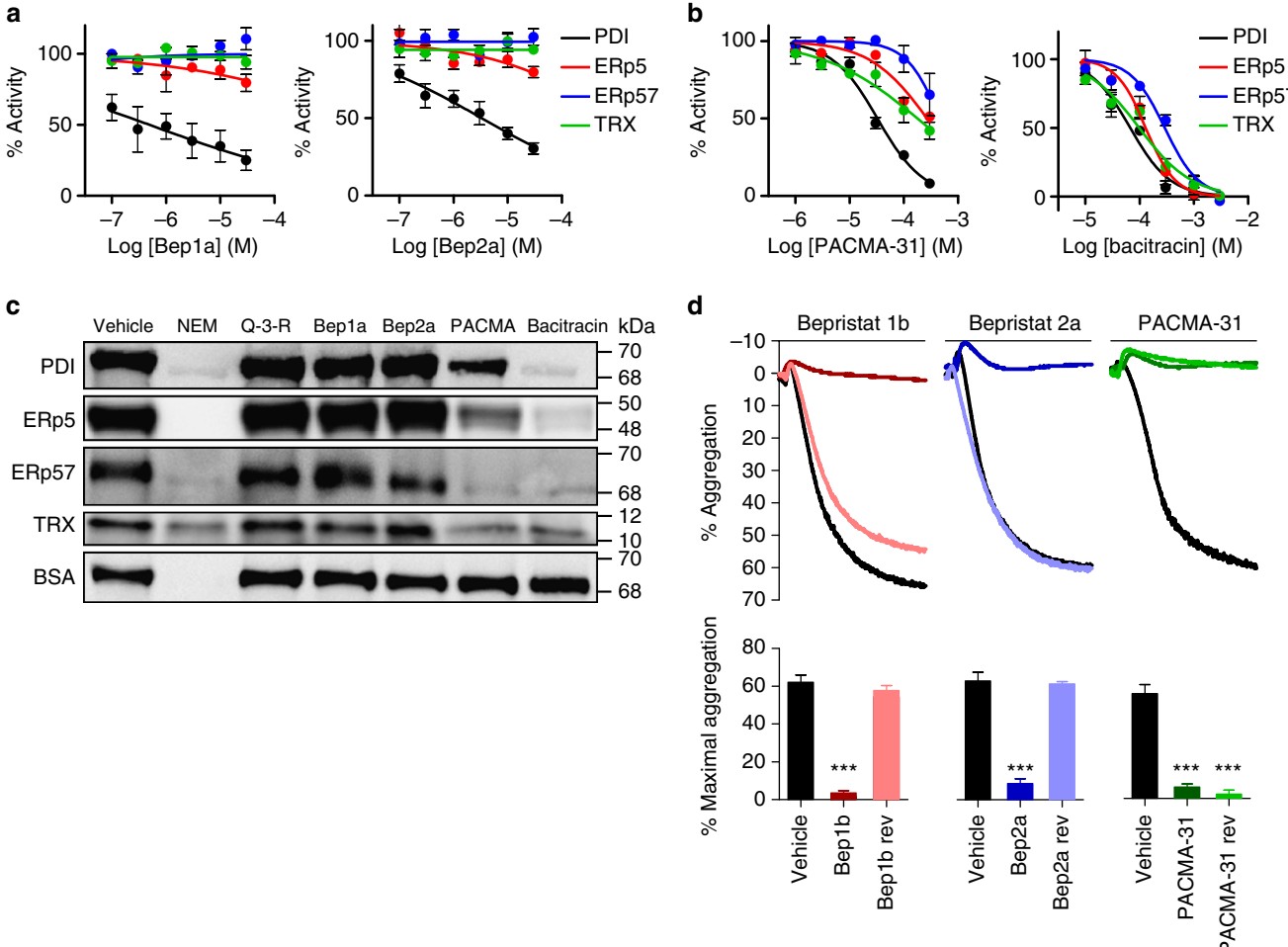

**Figure 2 | Selectivity and reversibility of bepristats.** The ability of (**a**) bepristat 1a (Bep1a) and bepristat 2a (Bep2a), or (**b**) PACMA-31 and bacitracin to inhibit the reductase activity of PDI (black), ERp5 (red), ERp57 (blue) and thioredoxin (TRX) (green) as measured in the insulin turbidimetric assay was evaluated. Values represent the per cent activity compared with samples exposed to vehicle alone ± s.e.m. (n = 2–3). (**c**) PDI, ERp5, ERp57, ERp72, TRX or BSA were incubated with vehicle, 300 μM N-ethylmaleimide (NEM), 150 μM quercetin-3-rutinoside (Q-3-R), 150 μM Bep1a, 150 μM Bep2a, 150 μM PACMA-31 or 150 μM bacitracin before exposure to MPB. Samples were then analysed by SDS–polyacrylamide gel electrophoresis, and MPB was detected by Cy5-labelled streptavidin. (**d**) Washed human platelets ($2 \times 10^8$ platelets per ml) were incubated with vehicle and stimulated with 3 μM SFLLRN (black tracings); incubated with either 30 μM Bep1b, 30 μM Bep2a or 30 μM PACMA-31, and then stimulated with 3 μM SFLLRN (dark tracings); or incubated with 30 μM Bep1b rev, 30 μM Bep2a rev or 30 μM PACMA-31, washed and subsequently stimulated with 3 μM SFLLRN (light tracings). Aggregation was monitored by light transmission aggregometry. Bar graphs represent the average maximal per cent aggregation ± s.e.m. (n = 3–7). One-way analysis of variance with Dunnett's post test: ***$P < 0.001$.

(Fig. 4a). In contrast, they both blocked activity of the **abb′** and **b′xa′** domains. PACMA-31 inhibited reductase activity of all PDI fragments in the insulin turbidimetric assay (Fig. 4a). These results demonstrate that bepristat 1a and bepristat 2a inhibit PDI reductase activity in the insulin turbidimetric assay by binding outside the catalytic motif, at **b′**.

The C-terminal end of the **b′** domain is connected to an x-linker that covers a deep hydrophobic pocket in **b′** and is thought to mediate the movement of the **a′** domain relative to the rest of the protein[39]. In a **b′x** fragment in which isoleucine 272 is mutated to alanine, the x-linker is constitutively associated with the hydrophobic patch on the **b′** domain[40]. 1-anilinonaphthalene-8-sulfonic acid (ANS) fluorescence was used to evaluate binding to hydrophobic regions on PDI in wild-type and mutant constructs. Binding of ANS to hydrophobic regions results in a marked increase in fluorescence when evaluated at $\lambda_{ex}$ 370 nm (Fig. 4b). ANS fluorescence was prominent on incubation with the isolated **b′x** domain and weak on incubation with isolated **a**, **a′** or **b** domains (Fig. 4b). ANS fluorescence was not observed on

incubation with I272A mutant (Supplementary Fig. 7), indicating that obstruction of binding pocket by the x-linker prevented ANS binding. Both bepristat 1a and bepristat 2a also interfered with the increase in ANS fluorescence observed on incubation with PDI (Fig. 4c). In contrast, incubation with PACMA-31 failed to block ANS fluorescence (Fig. 4c). Similar results were obtained when binding of ANS to the isolated **b′x** domain was evaluated (Fig. 4c).

**The x-linker is critical for augmenting PDI catalytic activity.**
Bepristats bind the **b′** domain and enhance catalytic activity at the **a** and **a′** domains, raising the question of how binding of a small molecule targeted to one domain modifies enzymatic activity at remote domains. To determine which domains contribute to the ability of bepristats to augment activity at the catalytic cysteines of PDI, the effect of bepristats on di-eosin-GSSG cleavage was tested using full-length PDI, **abb′** and **b′xa′**. Bepristats significantly augmented the catalytic activity of the **b′xa′** fragment (Fig. 5a). In contrast, bepristats failed to augment activity of the **abb′** fragment, which is missing the x-linker.

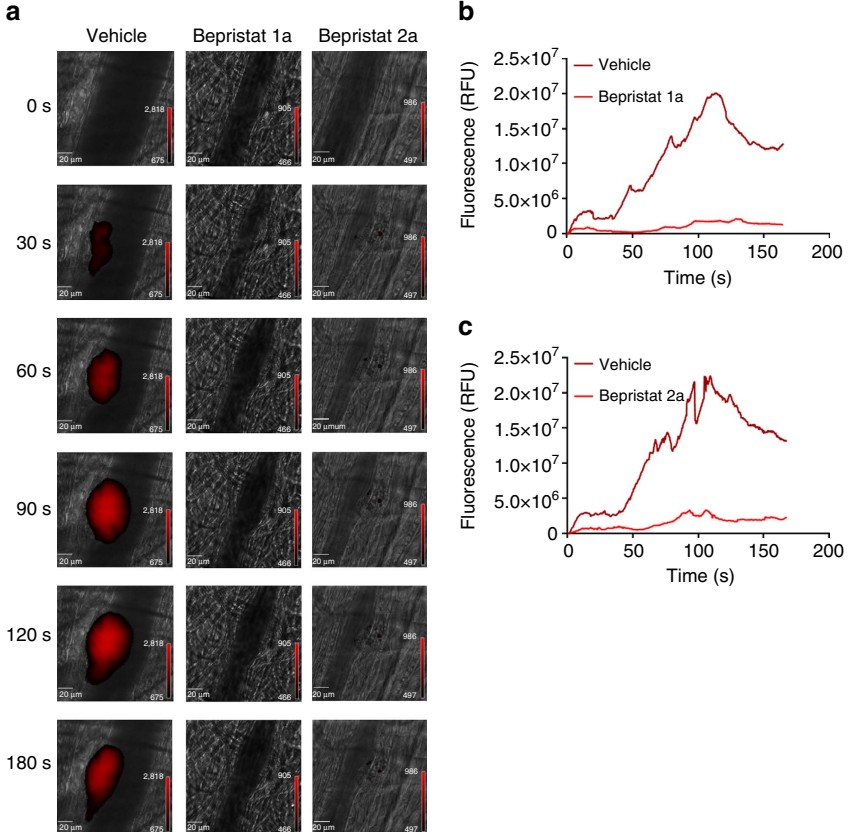

**Figure 3 | Bepristats inhibit thrombus formation following vascular injury.** (**a**) Platelet-specific anti-GPIbβ antibodies conjugated to Dylight 649 (0.1 μg per g body weight) were infused into mice. Mice were subsequently infused with either bepristat 1a (15 mg per kg body weight) or bepristat 2a (15 mg per kg body weight) as indicated. Thrombi were induced by laser injury of cremaster arterioles before ($n = 35$) and after ($n = 35$) infusion of bepristat 1a or before ($n = 42$) and after ($n = 28$) infusion of bepristat 2a. Thrombus formation was visualized by video microscopy for 180 s after injury. Fluorescence intensity scale bars, vehicle—888–7,884; bepristat 1a—466–905; bepristat 2a—497–986. Representative binarized images of platelets at the injury site before (vehicle), after bepristat 1a infusion (bepristat 1a) or after bepristat 2a infusion (bepristat 2a) are shown. Median integrated platelet-fluorescence intensity at the injury site in mice before (dark red) and after (light red) infusion of (**b**) bepristat 1a or (**c**) bepristat 2a infusion is plotted over time. RFU, relative fluorescence units. Scale bars, 20 μm.

The observation that bepristats target the same hydrophobic pocket on the **b′** domain that the x-linker associates with suggested that binding of bepristats results in displacement of the x-linker. To evaluate this possibility, we tested the protease sensitivity of **abb′x** in the presence and absence of bepristats. Proteinase K cleaves PDI from the C-terminal end[41]. Cleavage of **abb′x** by proteinase K occurred more rapidly in the presence of bepristat 1a and bepristat 2a than in their absence (Fig. 5b and Supplementary Fig. 8). The remaining fragment of proteinase K-treated **abb′x** had a molecular weight similar to **abb′**, consistent with the premise that proteinase K cleaves the x-linker. Bepristats did not interfere with proteinase K activity as evidenced by the fact that proteinase K cleaved ERp5 equally well in the presence or absence of bepristats. The effect of bepristats on movement of the x-linker was also evaluated using intrinsic fluorescence measurements. The x-linker includes a tryptophan residue (Trp-347) that associates with the hydrophobic binding pocket on the **b′** domain, resulting in an increase in intensity and a blue shift[41]. Incubation with bepristats resulted in a loss of intensity of intrinsic tryptophan fluorescence and a red shift (Fig. 5c), indicating that bepristats elicit movement of the x-linker on incubation with **b′x**. These results support a model whereby ligation of bepristats at the hydrophobic binding pocket results in displacement of the x-linker.

Small-angle X-ray scattering (SAXS) was used to determine whether the local effects of bepristats on interactions between the **b′** domain and the x-linker had global consequences on PDI conformation. PDI is a flexible protein that exhibits dynamic behaviour in aqueous solution. We used SAXS to measure the gyration radius (Rg) of PDI in aqueous solution. An overall molecular envelope of PDI was derived from these measurements. We found the gyration radius for full-length oxidized PDI to be 40.0 Å, whereas the gyration radius for reduced PDI was 34.0 Å. PDI complexed with bepristat 1b had a gyration radius of 35.3 Å and PDI complexed with bepristat 2b showed a gyration radius of 35.9 Å (Fig. 5d). These data suggest that bepristats constrain the dynamic behaviour of PDI. In the presence of bepristats, PDI adopts a more compact conformation that closely approximates the overall envelope of reduced PDI.

By binding the **b′** domain and eliciting a change in the global conformation in PDI, bepristats could modify disulfide bond formation at the CGHC motifs. Differential cysteine alkylation followed by mass spectroscopy under varying reduced glutathione (glutathione sulfhydryl, GSH):oxidized glutathione (glutathione disulfide, GSSG) ratios was used to quantify unpaired thiols and disulfide bonds within the CGHC motifs of the **a** and **a′** domains. This technique (Supplementary Fig. 9) uses $^{12}$C-IPA to label unpaired thiols. Disulfide bonds are then reduced using dithiothreitol (DTT) and the resulting free thiols are labelled using $^{13}$C-IPA. Differential cysteine alkylation showed that in a relatively oxidizing environment (high GSSG to GSH ratio), the fraction of reduced PDI was increased following incubation with

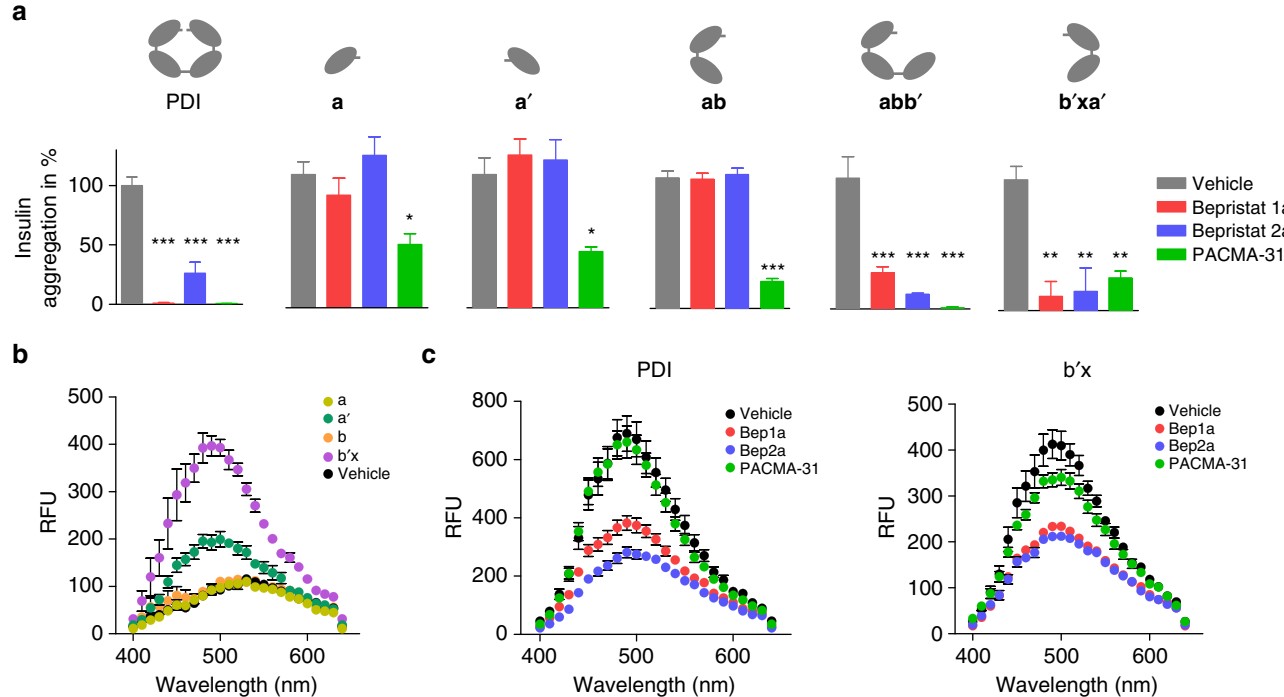

**Figure 4 | Bepristats associate with the substrate-binding pocket of the b′ domain.** (**a**) Evaluation of domain targets used either full-length PDI or PDI domains including **a** (residues 1–117), **a′** (residues 342–462), **ab** (residues 1-218), **abb′** (residues 1–331) or **b′xa′** (residues 219–462) as indicated. Full-length PDI or PDI domains were incubated with vehicle (grey), 30 μM bepristat 1a (red), 30 μM bepristat 2a (blue) or 100 μM PACMA-31 (green) for 30 min. Proteins were then assayed for activity in the insulin turbidimetric assay. Values represent per cent vehicle control ± s.e.m. ($n = 4$). One-way analysis of variance with Dunnett's post test: $*P < 0.05$; $**P < 0.01$; $***P < 0.001$. (**b**) Fluorescence monitored at $\lambda_{ex}$ 370 nm following incubation of 50 μM ANS with isolated **a** (gold), **a′** (green), **b** (orange), **b′x** (purple) or vehicle alone (black). (**c**) PDI or **b′x**, as indicated, was incubated in the presence of either vehicle (black), 100 μM bepristat 1a (red), 100 μM bepristat 2a (blue) or 100 μM PACMA-31 (green). Samples were then exposed to 50 μM ANS and fluorescence monitored following excitation at $\lambda_{ex}$ 370 nm. RFU, relative fluorescence units.

bepristats, as indicated by the baseline offset (Fig. 5e,f). This difference between samples exposed to bepristats and control samples is lost under reducing conditions (high GSH to GSSG ratio). Calculation of the redox potential of the two active sites demonstrated no difference between bepristat-exposed and control samples (Supplementary Table 1), since the effect of bepristats is overcome under more reducing conditions. However, these data show that the conformational change induced by bepristats impair active site disulfide bond formation under equilibrium conditions.

**Substrates stimulate thiol isomerase catalytic activity**. The bepristats affect PDI activity by a previously unrecognized mechanism involving engagement of a hydrophobic pocket on the **b′** domain with displacement of the x-linker resulting in a conformation change and enhanced reductase activity in the di-eosin-GSSG assay. Peptides such as mastoparan[42] have previously been shown to associate with the hydrophobic pocket on **b′**. Incubation of PDI with mastoparan enhanced the ability of PDI to cleave di-eosin-GSSG in a dose-dependent manner (Fig. 6a), even though mastoparan inhibits PDI activity in an RNase refolding assay[42]. Somatostatin, a 14-amino-acid peptide hormone, also binds PDI and stimulates cleavage of di-eosin-GSSG (Fig. 6b). In addition, a protein substrate of PDI, cathepsin G, elicits enhanced reductase activity in the di-eosin-GSSG assay (Fig. 6c). Like the bepristats, mastoparan, somatostatin and cathepsin G all decreased the apparent $K_M$ in the di-eosin-GSSG assay of PDI reductase activity (Fig. 6d). None of them altered the apparent $k_{cat}$.

Peptides and protein substrates associate with substrate-binding domains of other thiol isomerases[43]. To evaluate whether substrate-driven augmentation of catalytic activity is observed in other thiol isomerases, we tested the ability of mastoparan, somatostatin and cathepsin G to enhance cleavage of di-eosin-GSSG by ERp72, ERp57 and ERp5. All three substrates augmented the catalytic activity of ERp72 (Fig. 6a–c). In contrast, neither mastoparan, somatostatin nor cathepsin G was able to augment the catalytic activity of ERp57 or ERp5.

## Discussion

PDI family enzymes are multifunctional thiol isomerases that share a common ancestral thioredoxin-like domain. Gene duplication resulted in the formation of multidomain enzymes in eukaryotes in which some thioredoxin-like domains retained catalytic function while others evolved into substrate-binding domains[33]. While the catalytic domains within PDI are capable of autonomous activity, they demonstrate cooperative and enhanced behaviour within the context of full-length PDI[1,4,5]. The mechanism of this cooperativity, however, is poorly understood. We have identified an allosteric switch that modulates the catalytic activity of PDI. Our results indicate that binding of small molecules, peptides or proteins to the primary substrate-binding pocket on the **b′** domain displaces the x-linker from the hydrophobic pocket causing a conformational change that enhances catalytic activity of remote domains.

This mechanism was uncovered using newly identified small molecules, termed bepristats, that target the **b′** domain. The catalytic thioredoxin-like domains of PDI family proteins

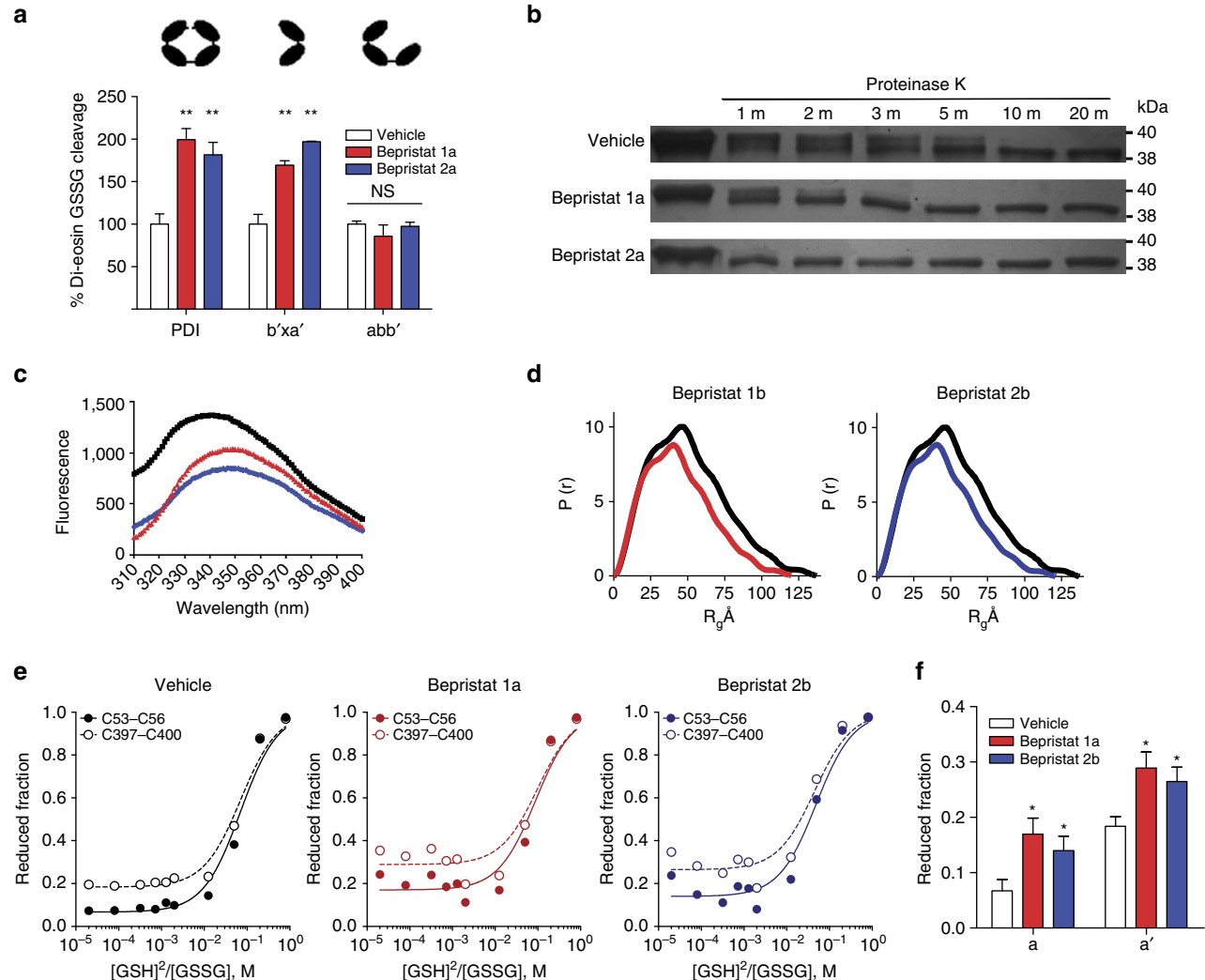

**Figure 5 | The x-linker is critical for bepristat-driven augmentation of PDI catalytic activity.** (**a**) Full-length PDI, **b′xa′**, and **abb′** were incubated with either vehicle (white) 30 μM bepristat 1a (red) or 30 μM bepristat 2a (blue) as indicated. PDI and PDI fragments were subsequently evaluated for activity in the di-eosin-GSSG assay. Values represent per cent vehicle control ± s.e.m. (n = 3–4). One-way analysis of variance with Dunnett's post test: *P < 0.05; **P < 0.01; NS, non-significant. (**b**) Detection of **abb′x** by silver staining following incubation with proteinase K for the indicated times in absence and presence of bepristat 1a and bepristat 2a. (**c**) Intrinsic fluorescence emission spectra of PDI in the absence (black) or presence of bepristat 1a (red) or bepristat 2a (blue). (**d**) SAXS profiles of PDI incubated in the presence of vehicle (black), bepristat 1b (red) or bepristat 2b (blue). (**e**) Plots of the ratio of reduced to oxidized PDI as a function of the ratio of GSH to GSSG in the absence (black) or presence of either 50 μM bepristat 1a (red) or 50 μM bepristat 2b (blue). The lines represent the best nonlinear least squares fit of the data. The calculated equilibrium constants were used to determine the standard redox potentials (Supplementary Fig. 9). Data points are the mean values from analysis of 2–4 peptides encompassing the active site cysteine residues. (**f**) Fraction of the active site dithiols/disulfides in the reduced state under oxidizing conditions in the absence (black) or presence of either 50 μM bepristat 1a (red) or 50 μM bepristat 2b (blue). The baseline offsets in the plots shown in **e** were a fitted parameter in the nonlinear least squares analysis. The error bars represent 1 s.e.

demonstrate a higher degree of homology than do the substrate-binding domains, which have evolved to associate with different substrate classes. As a result, compounds targeting catalytic domains generally lack selectivity among thiol isomerases. RL90, a monoclonal antibody that targets PDI, has been shown to cross-react with other closely related thiol isomerases, such as ERp57 (ref. 31). Similarly, we find that PACMA-31 is not entirely selective for PDI among thiol isomerases (Fig. 2). Although bepristats target the substrate-binding domain of PDI and elicit augmentation of di-eosin-GSSG cleavage, they demonstrate no such activity when tested against other thiol isomerases (Supplementary Fig. 1) and do not inhibit insulin aggregation mediated by ERp5, ERp57 or thioredoxin (Fig. 2). Bepristats were found to be active in <1% of several hundred assays tested, as

described in the PubChem database, attesting to their selectivity against a broader range of proteins. A second advantage of targeting the **b′** domain is reversibility. Compounds that target the catalytic domains tend to bind irreversibly via catalytic cysteines. By targeting the substrate-binding site, bepristats act as reversible inhibitors of PDI (Fig. 2 and Supplementary Fig. 5). Bepristats block platelet activation *in vitro* and impair platelet accumulation at sites of vascular injury in an *in vivo* model of thrombus formation (Fig. 3). These *in vivo* studies provide proof of principle for targeting the hydrophobic binding site of the **b′** domain of PDI in a clinical setting.

Bepristats are also useful in evaluating the role of the x-linker in modulating PDI activity. Protease digestion experiments and studies using the intrinsic fluorescence of Trp-347 to monitor

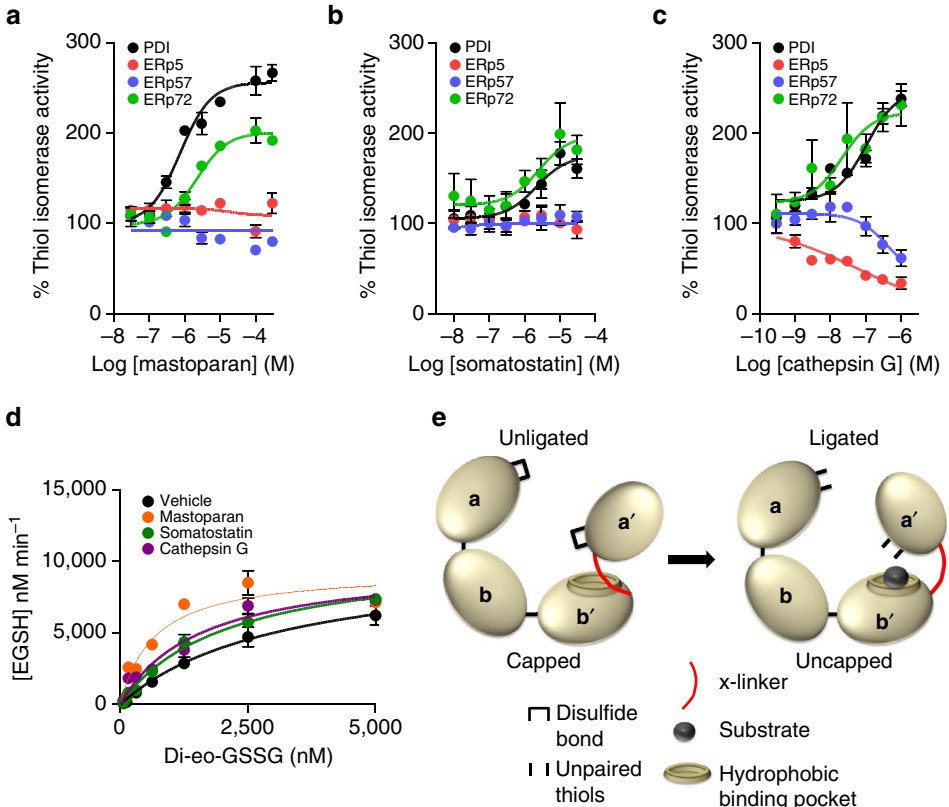

**Figure 6 | Protein substrates activate the allosteric switch mechanism in PDI and ERp72.** PDI (black), ERp5 (red), ERp57 (blue) or ERp72 (green) were incubated in the absence or presence of (**a**) mastoparan, (**b**) somatostatin or (**c**) cathepsin G at the indicated concentrations, and its effect on reductase activity monitored in the di-eosin-GSSG assay. (**d**) The effect of mastoparan, somatostatin and cathepsin G on $K_M$ and $k_{cat}$ was evaluated in the di-eosin-GSSG assay using increasing concentrations of di-eosin-GSSG. (**e**) Model of a substrate-driven allosteric switch that activates thiol isomerase catalytic activity. In the unligated state, the hydrophobic binding pocket within the **b**′ domain of PDI is capped and disulfide bond formation within the CGHC motif is favoured. Binding of substrate to the hydrophobic binding pocket results in displacement of the x-linker and the **a**′ domain, resulting in a more constrained conformation and favours unpaired cysteines within the CGHC motif. PDI reductase activity is enhanced in the ligated state.

movement of the x-linker confirmed displacement with bepristat exposure (Fig. 5). Displacement of the x-linker by bepristats is associated with a more constrained conformation, as demonstrated by SAXS. These studies indicate that binding of bepristats results in displacement of the x-linker and induces a conformational change in PDI. The net consequence appears to be a smaller binding pocket that cannot accommodate large substrates, and an **a**′-domain conformation that increases thiol-reductase activity for those substrates that can enter the smaller substrate-binding pocket.

While bepristats served as a convenient tool to evaluate this allosteric switch mechanism, peptides known to displace the x-linker demonstrated similar activity. Mastoparan and somatostatin both induced substantial augmentation of PDI-mediated di-eosin-GSSG cleavage (Fig. 6). Nuclear magnetic resonance spectroscopy showed that these peptides associate with the hydrophobic binding site on **b**′ that consists primarily of residues from α-helices 1 and 3, as well as from the core β-sheet[42,44]. Chemical shifts that occur on binding of either mastoparan or somatostatin have been mapped to hydrophobic residues adjacent to or within the substrate-binding pocket[42]. In the capped conformation of PDI, the x-linker binds this site. Peptide ligands such as mastoparan and somatostatin compete with and displace the x-linker, promoting an uncapped conformation[44]. The full range of substrates capable of augmenting PDI catalytic activity by associating with this binding pocket remains to be determined.

The observation that interactions at the hydrophobic binding pocket can influence the reductase activity at the CGHC motif

(Fig. 6d) demonstrates that PDI conformation is controlled in two distinct directions. In one direction, redox environment controls PDI conformation in a previously described mechanism that is initiated at the catalytic domains[41,45]. Reduction of the catalytic cystines in the CGHC motif is thought to trigger rotation of Trp-396, enabling it to interact with Arg-300 on the **b**′ domain, initiating a series of interactions at the **a**′–**b**′ interface that positions the **a**′ domain over the hydrophobic binding site on the **b**′ domain[41,45]. This constrained conformation is thought to be a means to limit substrate binding under reducing conditions[41]. Our results demonstrate a second mechanism controlling PDI conformation. This second mechanism is driven by substrate binding rather than by the redox environment. Binding of a substrate to the hydrophobic binding pocket results in displacement of the x-linker, an increase in active site cysteine thiolates and enhanced reduction of the substrate (Fig. 6e). This mechanism of substrate-driven augmentation of catalytic activity appears to be shared by some thiol isomerases, but not by others. Human ERp72 demonstrates augmentation of catalytic activity on binding mastoparan, somatostatin or cathepsin G while ERp57 and ERp5 do not (Fig. 6). Further studies will be required to identify the extent that a substrate-driven allosteric switch mechanism is used among thiol isomerases.

## Methods
**Protein purification.** Recombinant 'double-tagged' (streptavidin-binding protein-tagged and FLAG-tagged) full-length PDI, ERp57, recombinant His-tagged full-length ERp5, ERp72 and PDI domain fragments were cloned into a pET-15b

vector at the NdeI and BamHI sites and transformed into *Escherichia coli* Origami B (DE3) cells (EMD Chemicals). The recombinant proteins were expressed and isolated by affinity chromatography with complete His-Tag purification resin (Roche Applied Science) or Pierce High Capacity Streptavidin Agarose beads and purified on a Superdex 200 (GE Healthcare). Human thioredoxin was purchased from Abcam.

**Small-angle x-ray scattering.** Further purification of full-length PDI was achieved by gel filtration. PDI (1.5, 3.0 and 4.5 mg ml$^{-1}$) was dialysed against 20 mM Tris, 150 mM NaCl, 5% glycerol (pH: 8.0) containing 0.5 mM bepristat 1b or 2b or dimethylsulfoxide control at 4 °C overnight. Evaluation by SAXS was performed on the SIBYLS beamline in the Advanced Light Source using a high-throughput data collection method.

**Redox potential determination.** The redox potentials of the **a** (Cys53-Cys56) and **a'** (Cys397-Cys400) active-site dithiols/disulfides of human wild-type PDI in the absence or presence of bepristat 1a or bepristat 2b were determined by differential cysteine alkylation and mass spectrometry. Recombinant PDI (5 μM) was incubated in the absence or presence of bepristat 1a or bepristat 2b (50 μM) in argon-flushed phosphate-buffered saline containing 0.1 mM EDTA, 0.2 mM oxidized glutathione (GSSG, Sigma) and indicated concentrations of reduced glutathione (GSH, Sigma) for 18 h at room temperature. Microcentrifuge tubes were flushed with argon before sealing to prevent oxidation by ambient air during the incubation period. Unpaired cysteine thiols in PDI and mutants were alkylated with 5 mM 2-iodo-*N*-phenylacetamide ($^{12}$C-IPA, Cambridge Isotopes) for 1 h at room temperature. The proteins were resolved on SDS–PAGE and stained with SYPRO Ruby (Invitrogen). The PDI bands were excised, destained, dried, incubated with 100 mM DTT and washed. The fully reduced proteins were alkylated with 5 mM 2-iodo-*N*-phenylacetamide where all six carbon atoms of the phenyl ring have a mass of 13 ($^{13}$C-IPA) (Cambridge Isotopes). The gel slices were washed and dried before digestion of proteins with 12 ng μl$^{-1}$ of chymotrypsin (Roche) in 25 mM NH$_4$CO$_2$ overnight at 25 °C. Peptides were eluted from the slices with 5% formic acid and 50% acetonitrile. Liquid chromatography, mass spectrometry and data analysis were performed as described[46].

The fraction of reduced active-site disulfide bond was measured from the relative ion abundance of peptides containing $^{12}$C-IPA and $^{13}$C-IPA. To calculate ion abundance of peptides, extracted ion chromatograms were generated using the XCalibur Qual Browser software (v2.1.0; Thermo Scientific). The area was calculated using the automated peak detection function built into the software. The ratio of $^{12}$C-IPA and $^{13}$C-IPA alkylation represents the fraction of the cysteine in the population that is in the reduced state. The results were expressed as the ratio of reduced to oxidized protein and fitted to equation 1:

$$R = \frac{B + \left\{ (1 - B) \times \left( \frac{[GSH]^2}{[GSSG]} \right) \right\}}{K_{eq} + \left( \frac{[GSH]^2}{[GSSG]} \right)} \qquad (1)$$

where $R$ is the fraction of reduced protein at equilibrium, $B$ is the baseline fraction of the cysteine in the population that is in the reduced state and $K_{eq}$ is the equilibrium constant. The standard redox potential ($E^{0\prime}$) of the PDI active-site disulfides were calculated using the Nernst equation (equation (2)):

$$E^{0\prime} = E^{0\prime}_{GSSG} - \frac{RT}{2F} \ln K_{eq} \qquad (2)$$

using a value of −240 mV for the standard redox potential of the GSSG disulfide bond.

***In vivo* experiment.** All protocols involving the use of animals were in compliance with the National Institutes of Healths Guide for the Care and the Beth Israel Deaconess Medical Center Institutional Animal Care and Use. Intravital video microscopy of the cremaster muscle microcirculation was performed, and digital images were captured with an Orca Flash 4.0v2 sCMOS camera (Hamamatsu Photonics K.K., Shizuoka Pref., Japan)[22,47]. Representative images are presented, but the median curves include the full data. The kinetics of platelet thrombus formation were analysed by determining median fluorescence values over time in ∼30–40 thrombi in three mice (all 8-week-old males, C57/BL6J background).

Two hours before surgery, 100 mg kg$^{-1}$ of the suicide P450 inhibitor 1-aminobenzotriazole (ABT) was administered intraperitoneally to each mouse. The cremaster muscle was then surgically exposed. Before arteriolar wall injury, DyLight-labelled antibodies were infused intravenously together with bepristats or vehicle control. Injury to a cremaster arteriolar vessel (30–50 μm diameter) was induced with a MicroPoint laser system, Andor Technology, Ltd., Belfast Ireland) focused through the microscope objective, parfocal with the focal plane and tuned to 440 nm through a dye cell containing 5 mM coumarin in methanol. Data were captured digitally from two fluorescence channels, 488/520 nm and 647/670 nm, as well as a brightfield channel. Data acquisition was initiated both before and following an ablation laser pulse for each injury. The microscope system was controlled and images were collected and analysed using SlideBook 6.0 (Intelligent Imaging Innovations, Denver, CO).

Before induction of the thrombus but after injection of the fluorescently labelled antibody, ∼30 time points were recorded. Subsequently, a thrombus is initiated and recorded for ∼170 s. Post capture, an upstream region is defined near the site of each thrombus. The maximum pixel intensity in this region is extracted for each time point. The mean of maximal intensity values in the upstream region for each frame is calculated and used as the threshold to define those pixels containing signal. Extracting the values of these pixels and summing them, we obtained the uncorrected integrated intensity for each time point. The area of this region (in pixels) is also reported. Using this information, the actual integrated intensity for each frame is calculated according to the following formula:

actual integrated intensity = (sum of the uncorrected integrated intensity) − (mean of the maxima from the upstream region × area of the uncorrected integrated intensity).

**Tryptophan fluorescence.** Intrinsic fluorescence spectra were performed in a reaction volume of 50 μl with 5 μM of PDI in 50 mM Tris-HCl buffer containing 150 mM NaCl (pH 7.6). Emission spectra were recorded at 310–400 nm with excitation at 290 nm. Bepristats were used at a concentration of 50 μM.

**MPB binding.** MPB-binding experiments were performed in a reaction volume of 37.5 μl with 5 μM of thiol isomerase or bovine serum albumin (BSA) in Tris-buffered saline (TBS), in the presence and absence of 1 mM of *N*-ethylmaleimide or bacitracin, and 150 μM of the mentioned other inhibitors. The reaction mixture was incubated at 37 °C for 1 h. Subsequently, the reaction mixtures were incubated with 25 μM of MPB. The labelling was performed for 20 min at 25 °C. A total of 12.5 μl of 4 × Laemmli Sample Buffer with 5% β-mercaptoethanol was added to each of the samples, followed by heating at 95 °C for 10 min. From each sample, 10 μl was loaded on a 12% SDS–PAGE gel, followed by transfer on a nitrocellulose membrane. After blocking the membrane with TBS-T containing 5% BSA for 1 h, the membrane was incubated with a 1:2,000 dilution of Cy5-tagged streptavidin for an hour in the dark. Detection of the immunoblots was performed using ImageQuant LAS 4000, and analysis was performed using ImageQuant TL software.

**1-Anilinonaphthalene-8-sulfonic acid fluorescence.** The binding of 1-anilino-naphthalene-8-sulfonic acid (ANS) to full-length PDI and the different domains of PDI was assessed by incubating 5 μM of protein in the presence or absence of 100 μM of the indicated inhibitors in 175 μl of TBS at 37 °C for 1 h. Subsequently, 50 μM of ANS was added and the mixture was incubated in the dark at 25 °C for 20 min. Fluorescence spectrum (Ex: 370 nm, Em: 400–700 nm) was measured in a 384-well plate. The experiment was performed in triplicate.

**Platelet aggregation.** Platelet-rich plasma was obtained from healthy volunteers. Informed consent was obtained from all volunteers. Platelets were isolated by centrifugation at 2,000*g* and resuspended in Hepes-Tyrode buffer (134 mM NaCl, 0.34 mM sodium phosphate, 2.9 mM KCl, 12 mM sodium bicarbonate, 20 mM HEPES, 1 mM magnesium chloride, 5 mM glucose (pH 7.3)). Washed human platelets (2.5 × 10$^8$ platelets per ml) were incubated with the indicated concentrations of bepristats and PACMA-31 at 37 °C for 20 min and then exposed to 3 μM PAR-1-activating peptide SFLLRN. Aggregation was measured using a Chrono-Log 680 Aggregation System.

**Insulin turbidimetric assay.** The thiol isomerase-catalysed reduction of insulin was assayed by measuring the increase in turbidity as detected at an optical density of 650 nm using a Spectramax M3 (Molecular Devices, Sunnyvale, CA). The validation assay consisted of 175 nM of PDI in a solution containing 100 mM potassium phosphate (pH 7.4) containing 0.2 mM bovine insulin, 2 mM EDTA and 0.3 mM DTT (all purchased from Sigma Aldrich, St Louis, MO), inhibitors were used at the concentrations indicated. The reaction was performed at 25 °C for 1 h and 30 min. For assays of thiol isomerase selectivity,11 nM PDI, ERp57 or thioredoxin, or 33 nM ERp5 were assayed in similar buffer conditions as described above and inhibitors were used at the indicated concentration. The reaction was performed at 37 °C for 45 min. For assays of isolated domains studies, 400 nM protein was used, except for **a**, **a'** and **ab**, in which 800 nM protein was used. The assay was performed in similar buffer conditions as described above. Bepristat 1a, 2a and PACMA-31 were used at a final concentration of 15 μM in these studies. For calculation of the IC$_{50}$, we used absorbance values recorded at 25 min after insulin aggregation is first detectable in the vehicle control compared with the corresponding value at that time point for specimens containing the study drug.

Reversibility studies were performed by incubating 20 μM PDI with 3 μM of bepristat 1a, 6 μM of bepristat 2a or 300 μM of PACMA-31. After 30 min equilibration, the PDI-inhibitor mixture was diluted 100-fold into the above-mentioned assay buffer. The ability to reduce insulin of these mixtures was compared with PDI in the presence or absence of 3 μM bepristat 1a or 0.03 μM bepristat 1a, 6 μM bepristat 2a or 0.06 μM bepristat 2a or 300 μM PACMA-31 or 3 μM PACMA-31.

**Di-eosin-GSSG disulfide reductase assay.** The probe di-eosin glutathione disulfide, di-eosin- GSSG, was prepared incubating 100 μM of GSSG with 1 mM of

eosin isothiocyanate in 100 mM potassium phosphate containing 2 mM EDTA (pH 7.4), at 25 °C overnight in the dark[36]. The mixture was passed down a PD-10 column and different fractions were collected. A fold change of fluorescence (Ex: 520 nm, Em: 550 nm) was calculated using samples subjected to either vehicle or 20 mM of DTT. Any fraction with a fold change >5 was kept. Using a stock of 10 mM eosin isothiocyanate a standard curve was generated and concentration of samples were determined accordingly. Di-eosin-GSSG cleavage of purified thiol isomerases and PDI domains was monitored in a 96-well fluorescence plate format. PDI, AGHA-PDI, ERp5, ERp57, ERp72 and PDI domains were assayed at 50 nM in the presence or absence of the indicated small molecules, peptides or cathepsin G. The assay included 100 mM potassium phosphate (pH 7.4) containing 2 mM EDTA, 5 μM DTT and 150 nM of the di-eosin-GSSG probe. The increase in fluorescence was determined for 20 min by excitation at 520 nm and emission at 550 nm in a Synergy Biotek 4. The reduction of 150 nM di-eosin-GSSG by 5 μM DTT in the presence or absence of mentioned small molecules, peptides or proteins served as a negative control.

Michaelis–Menten analysis was assayed using PDI at 20 nM and small molecules, peptides or cathepsin G at concentrations causing maximal augmentation under similar buffer conditions as described above. Eight-point response curves were generated using a range of different di-eosin-GSSG concentrations (50–5,000 nM). Enzyme kinetics analysis was performed using Graphpad Prism 5.0.

**Proteolysis assay.** Proteinase K (2 μg ml$^{-1}$) was incubated with 1.25 μg of the abb′x fragment in 50 mM Tris-HCl containing 5 mM CaCl$_2$ and 10 mM DTT. Reactions were aborted with the addition of 0.5 mM phenylmethanesulfonyl fluoride. Subsequently, samples were subjected to Laemmli Sample buffer with 5% β-mercaptoethanol and heated at 95 °C for 10 min. Each sample was loaded on a 12% SDS–PAGE gel, followed by silver staining using the Pierce Silver Stain Kit (Thermo Scientific).

**Data availability.** Data supporting the findings of this study are available within the article (and its Supplementary Information files) and from the corresponding author on reasonable request.

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

## Acknowledgements

This work was funded by grants from NHLBI (U54 HL112302, R01 HL125275, R01 HL112809, T32 HL007917, T32 HL16324-02, and HL116324), NIDA (DA032476), NIH-MLPCN programme (U54 HG005032), and the Hemostasis and Thrombosis Research Society. The authors are grateful for the SAXS studies conducted at the Advanced Light Source (ALS), a national user facility operated by Lawrence Berkeley National Laboratory on behalf of the Department of Energy, Office of Basic Energy Sciences, through the Integrated Diffraction Analysis Technologies (IDAT) programme, supported by DOE Office of Biological and Environmental Research. Additional support comes from the National Institutes of Health project MINOS (R01GM105404).

## Author contributions

R.H.B., P.B., L.L., A.F. and D.R.K. performed functional assays of thiol isomerase activity; P.P.N. and J.P. performed chemical synthesis; R.H.B., P.B. and L.L. performed thiol isomerase production and characterization; P.B. performed intravital microscopy; L.L. performed intrinsic fluorescence experiments; R.H.B performed platelet function studies and proteinase digestions; J.C. and K.M.C. performed and P.J.H. supervised differential alkylation followed by mass spectroscopy; M.H. performed computational modelling and interpreted SAXS data; R.F. conceived the project; R.H.B. and R.F. prepared the manuscript; R.H.B., P.B., B.F., P.J.H., P.P.N. and R.F. edited the manuscript.

## Additional information

**Competing financial interests**: R.F. and P.P.N. are co-inventors on pending patents describing bepristats 1 and 2. The remaining authors declare no competing financial interests.

**How to cite this article**: Bekendam, R. H. *et al.* A substrate-driven allosteric switch that enhances PDI catalytic activity. *Nat. Commun.* 7:12579 doi: 10.1038/ncomms12579 (2016).

