## [Peer Review File · Nature Communications]

Reviewer #1 (Remarks to the Author):

Protein disulfide isomerase (PDI) is implicated in thrombosis and clot formation. Inhibitors of PDI can block thrombosis. PDI is a thiol isomerase consisting of four thioredoxin-like domains arranged in a horse shoe shape structure. The a and a' domains mediate disulfide bond shuffling and the b and b' domains substrate binding. In this manuscript the authors have identified a new class of compounds called bepristats that target a substrate binding pocket b'. Through careful analysis of activity assays using two substrates; namely insulin and diosin glutathione disulfide, they find that the new inhibitors reveal a substrate driven allosteric switch that enhances PDI catalytic activity. The new inhibitors also block thrombosis. Over all, this manuscript is of high class and deserves to be published.

The manuscript presents high class experimental results, which are performed with a range of methods and generally well described. I have some ideas that may improve the manuscript which the authors should consider.

1. In the "Summary" on page 2, it is suggested that the authors in the first sentence add that PDI is also present on cell surface and secreted. The second sentence contains a cryptic domain structure with an x, which is left unexplained. Even the "Introduction" has a problem with the x, which is not explained. It is suggested that the authors write: The thioredoxin-like domain structure of PDI is a, b, b' etc, but also explain, that x is an x-linker (as well described on page 8, second paragraph). Adding thioredoxin-like gives an explanation and also what x is important for the reader. The x is actually a very important part of the manuscript.
2. The same points as raised above, before the "Summary", can be applied to the "Introduction". To use in thioredoxin-like and explaining what x is. In fact, the x linker comes abruptly not until on page 4.
3. On page 6, line 7, the turbidimetric assay to assess their ability to inhibit ERp5, ERp57 and thioredoxin activity is incorrect. As it now reads, it looks like thioredoxin reductase activity, which is something different from thioredoxin. Either skip the reductase or move it further up in the sentence.
4. Regarding thioredoxin, which source of thioredoxin was used? Is it human or E. coli? On page 6, later; TxR is written on line 10. What is that, is it a misspelling on Trx or what?
5. In "Supplemental Methods" on page 3, end of first paragraph. Wright: is done a redox potential of glutathione, rather than GSSG disulfide bond. Previously on page 2, oxidized glutathione is not formally correct, but glutathione disulfide is.

Reviewer #2 (Remarks to the Author):

A. In their paper, the authors identify novel PDI inhibitors named bepristats that inhibit PDI activity in the insulin reductase assay and in contrast, enhanced PDI reductase activity in the di-e-GSSG assay. Their inhibitory effect was reversible and selective for PDI and was also shown in vivo in mice model of thrombosis. The authors next showed that bepristats associate with the hydrophobic pocket in the b' domain of PDI, displacing the x-linker that covers it and induce a conformational change that enhance PDI reductase activity.

B. This paper adds an important and novel data on the mechanism of PDI activity regulation showing that substrate binding to the b' domain of PDI can induce allosteric change in the catalytic domain of the enzyme.

C. The data and methodology used are of good quality and approach. The presentation is nice but needs some corrections and improvements as detailed in point F.

D. The statistics used is appropriate.

E. The conclusion are quiet robust and reliable. It needs restrictions and clarifications in some points as detailed in point F.

F. The following improvements and remarks should be addressed:

Experiments:

The aggregometry results shown in Fig 2d demonstrate that bepristats inhibit platelet aggregation. The authors need to show that this inhibitory effect is not due to inhibition of platelets activation and subsequent PDI secretion, for example by showing that other platelets activation markers (such as granule secretion) are not affected by these compounds.

General remarks:

Results and figures

1. In page 5 and fig. 1, the specific time point or slope measured in the insulin reductase assay needs to be clarified and mentioned in the text and figure legend.
2. Page 5, line 11: where do the authors show that bepristats do not affect di-e-GSSG cleavage in the absence of PDI?
3. Fig 1C is unclear, Please provide more precise explanation in the figure legend. More detailed information is also required for Fig 1D.
4. Page 6: the term "vascular thiol isomerases" needs to be mentioned and detailed in the introduction section.
5. Page 6 and Suppl. S2: The use of NEM as a control for irreversible binding of thiols should be explained in the text.
6. In page 6, please explain why did you use bepristat 1b with enhanced esterase resistance.
7. Page 7 and suppl. S3, the authors needs to address the incomplete reversibility of PACMA-1.
8. Page 7, please provide references in the text to the suppl. Videos.
9. Page 7 and Fig 3: the term "vehicle" is not provided in the methods. Only control before adding bepristats is mentioned. Please clarify this inconsistency.
10. Page 7, correct "by 14.9%" to "to 14.9%".
11. Figure 5: In the B panel, please provide the sizes of the proteins in the gel and explain what exactly shown there in this picture in the text (page 9). In D panel, one can see only the red line; the blue line should be shown as well even though these lines merge. In the F panel, explain more clearly in the legend what is shown, at what ratio of the GSH/GSSG this fraction is measured? Is it a bar representation of one of the points shown in panel E?.
12. Page 11, the last line should be written more accurately. Only Cathepsin inhibited the catalytic activity of ERp5 and Erp57 according to Fig 6.
13. Suppl. Fig S1: the Y title should be corrected to "PDI activity in %"
14. Suppl. Fig S2: what are the numbers (BRD1035, BRD4832)? It was not mentioned anywhere before - it should be consistent. The statistical significance should be shown in the graphs.
15. Suppl. Fig S4: this figure has no reference and explanation in the text. It can be either removed or be explained in the results and figure legend.
16. Suppl. Fig S5: In the graph instead of "ANS" it should be "vehicle"

Discussion

1. In line 10 of the second paragraph the term "reductase activity" should be more specific. It was only inhibited in the insulin assay for these enzymes and not in the di-e-GSSG assay. The authors need to more clearly discriminate between these assays throughout the whole manuscript because these assays display opposite results and this can be confusing to the reader.
 2. Line 11 of the same paragraph is not clear and need to be clarified.
 3. Page 13, line 1 of the second paragraph: the word "also" appears twice. Erase the first one.
 4. Page 13: The authors should address the finding that bepristats inhibited only PDI, while the other peptides also inhibit ERp57 and provide possible explanation for that.
 5. Page 13, last line: the fig should be 6d instead of 5d.
- G. References are appropriate.
- H. Abstract and conclusions are appropriate, except for some issues that should be addressed in the "Discussion" as detailed in point F. The introduction needs some improvement. It should contain more details about the other thiol isomerases mentioned in the results and what is known about other peptides that bind the b' domain like mastoparan. These peptides are firstly mentioned in the results section even though they present the same binding pattern as the bepristats. In general, the authors should go through the results section and provide some of the previously known data that is first mentioned there in the introduction section instead.

Reviewer #3 (Remarks to the Author):

PDI has been increasingly implicated in the initiation and progression of many pathological events.

The novel discovery of a new class of small molecular weight (SMW) PDI-specific allosteric inhibitors reported in this manuscript, is very timely and should be widely communicated.

The inhibitors described in this study, bepristats (BPS), were identified through the screening of ~350K SMW compounds.

The most potent of these BPS1 and BPS2 are the focus of the study described in the MS. BPS 1 & 2 inhibited the PDI isomerase activity monitored via the insulin-turbidity assay and in vitro platelet aggregation. Interestingly, the BPSs stimulated the PDI-catalyzed reduction of the SMW fluorogenic pseudo-substrate, di-eosin-GSSG. In addition, the authors demonstrated that these inhibitory/stimulatory effects were reversible.

The effectiveness of BPSs in inhibiting thrombus formation was also demonstrated in vivo via intravital microscopy.

In an attempt to identify the PDI-interaction domains of the BPSs, various active site domains (a, a') {plus minus} the b, b' and x domains were tested to BPS-catalytic effects. These studies indicated that the BPSs were interacting with the b' domain BUT that the x-linker domain was essential for the stimulation of the catalytic activity. This hypothesis was validated by SAXS data showing that BPS binding resulted in large conformational changes that affected the gyration radius of PDI, resulting in a more compact conformation for the enzyme. The net consequence of this appears to be a smaller binding pocket that can't accommodate large substrates, and an a'-domain conformation that increases thiol-reductase activity for those substrates that can enter the smaller substrate binding pocket.

The authors also demonstrated that other peptides besides BPSs can also affect a similar conformational change in PDI and that the BPSs are novel in that they can only affect these conf changes in PDI and not in other PDI-like proteins.

The data is solid and fit well with the interpretations /conclusions of the authors.

The MS is clearly written will be easily understood by a general audience.

Suggestion: the authors should do a better job of explaining the how the observed and hypothesized structural effects of the BPSs can be rationalized with both the inhibition thrombus and insulin turbidity and paradoxically the activation of reductase activation. Perhaps they can make use of this reviewers comments above.

Minor point: Fig 2a-black line is missing in the legend for PDI.

In summary, this is a very novel finding befitting the targeted journal and it should be communicated with haste.

Beth Israel Deaconess
Medical Center

Harvard Medical School

Division of
Hemostasis &
Thrombosis

Center for
Hemostasis,
Thrombosis and
Vascular Biology

Bruce Furie MD
Chief, Division of
Hemostasis &
Thrombosis
Professor
Harvard Medical
School

Kenneth A. Bauer
MD
Professor
Harvard Medical
School

Rob Flaumenhaft
MD PhD
Associate Professor
Harvard Medical
School

Mingdong Huang
PhD
Lecturer
Harvard Medical
School

Jeffrey I. Zwicker
MD
Associate Professor
Harvard Medical
School

Natalia Beglova
PhD
Assistant Professor
Harvard Medical
School

Dear Dr. Pastore,

May 19, 2016

We thank you and the reviewers for the thoughtful critique of our manuscript. We especially appreciated how all three reviewers provided useful suggestions to improve the manuscript while also providing encouraging and complimentary commentary on the significance of the results and rigor of the approaches used. Overall, we feel that the critique has strengthened the manuscript significantly. Our point-by-point responses to the reviewers are below and changes to the text are denoted in red throughout the manuscript.

Reviewer #1:

1. *In the "Summary" on page 2, it is suggested that the authors in the first sentence add that PDI is also present on cell surface and secreted. The second sentence contains a cryptic domain structure with an x, which is left unexplained. Even the "Introduction" has a problem with the x, which is not explained. It is suggested that the authors write: The thioredoxin-like domain structure of PDI is a, b, b' etc, but also explain, that x is an x-linker (as well described on page 8, second paragraph). Adding thioredoxin-like gives an explanation and also what x is important for the reader. The x is actually a very important part of the manuscript.*

We thank the reviewer for these insightful suggestions and agree that the x-linker deserves more explanation earlier in the manuscript. We have adapted the Summary accordingly. Word limit restrictions prevented us from including the localization of PDI to cell surfaces and its secretion of cells in the Summary. However, we have now included this information in the Introduction.
(Summary, lines 2-4; Introduction, lines 3-5)

2. *The same points as raised above, before the "Summary", can be applied to the "Introduction". To use in thioredoxin-like and explaining what x is. In fact, the x linker comes abruptly not until on page 4.*

The reviewer's point is well taken. The introduction has been revised accordingly.
(Introduction, paragraph 1, starting at line 8)

3. *On page 6, line 7, the turbidimetric assay to assess their ability to inhibit ERp5, ERp57 and thioredoxin activity is incorrect. As it now reads, it looks like thioredoxin reductase activity, which is something different from thioredoxin. Either skip the reductase or move it further up in the sentence.*

We agree that this language is ambiguous and have changed the phrasing to improve clarity. (Page 6, Line 4 of 1st full paragraph)

4. *Regarding thioredoxin, which source of thioredoxin was used? Is it human or E. coli? On page 6, later; TxR is written on line 10. What is that, is it a misspelling on Trx or what?*

The source of the thioredoxin was human. This information has been added to the Supplemental Methods section. "TxR" has been changed to thioredoxin. (Page 6, line 8 of 1st full paragraph)

5. In "Supplemental Methods" on page 3, end of first paragraph. Wright: is done a redox potential of glutathione, rather than GSSG disulfide bond. Previously on page 2, oxidized glutathione is not formally correct, but glutathione disulfide is.

This is an important point. At the first reference to GSSG and GSH in the manuscript we refer to them as 'oxidized glutathione' and 'reduced glutathione' respectively. In the explanation within brackets, we added glutathione disulfide and glutathione sulfhydryl. (Page 10, line 3 of last paragraph)

Reviewer #2:

Experiments:

The aggregometry results shown in Fig 2d demonstrate that bepristats inhibit platelet aggregation. The authors need to show that this inhibitory effect is not due to inhibition of platelets activation and subsequent PDI secretion, for example by showing that other platelets activation markers (such as granule secretion) are not affected by these compounds.

This is an excellent point. We have added new data to the manuscript as 'Supplementary figure S4' showing the effect of bepristats on P-Selectin expression. Under conditions in which bepristats inhibit platelet aggregation, we observed no significant effect of bepristats on P-selectin expression.

General remarks:

Results and figures

1. In page 5 and fig. 1, the specific time point or slope measured in the insulin reductase assay needs to be clarified and mentioned in the text and figure legend.

We thank the reviewer for this well-taken point. We have provided in the 'Supplementary Methods' section a more detailed explanation of our approach to analyzing data from this assay (Supplementary Materials, Page 6, First paragraph of 'Insulin Turbidimetric Assay', last sentence) and have supplied additional detail in the figure legend as well (Page 20, Figure 1, third and fourth sentence).

2. Page 5, line 11: where do the authors show that bepristats do not affect di-e-GSSG cleavage in the absence of PDI?

We thank the reviewer for this comment. To address the possibility of a direct effect of the bepristats on the di-eosin-GSSG probe, we included an experiment in which the bepristats did not demonstrate a significant impact on di-eosin-GSSG fluorescence in the presence of a catalytically dead variant of PDI (i.e., the AGHA mutant; Figure 1C). In order to provide further support for this observation, a figure depicting an additional control experiment including probe with no bepristat has been added to 'Supplementary Figure S1.'

3. Fig 1C is unclear, Please provide more precise explanation in the figure legend. More detailed information is also required for Fig 1D.

Figure legends for both panels (1C and 1D) have been updated to improve clarity.

4. Page 6: the term "vascular thiol isomerases" needs to be mentioned and detailed in the introduction section.

The manuscript has been revised such that this term is now defined. (Introduction, page 4, end of first paragraph)

5. *Page 6 and Suppl. S2: The use of NEM as a control for irreversible binding of thiols should be explained in the text.*

We thank the reviewer for this insightful comment. The text has been modified to explain the use of NEM as a control for nonspecific MPB labeling of PDI. (Page 6, second paragraph)

6. *In page 6, please explain why did you use bepristat 1b with enhanced esterase resistance.*

Platelets contain intrinsic esterases that could interfere with the activity of bepristat 1a. Because the ester group in bepristat 1b is chemically protected, inhibition of platelet aggregation can be achieved with this compound. We have revised the manuscript to include this explanation. (Page 7, lines 4-5)

7. *Page 7 and suppl. S3, the authors needs to address the incomplete reversibility of PACMA-1.*

Indeed, partial reversibility of PACMA-31 was observed in this assay, though these data still indicate that PACMA-31 inhibition is largely irreversible. In the initial paper describing PACMA-31 (Xu, et al. PNAS 2012), several techniques were used to show that PACMA-31 is an irreversible PDI inhibitor (fluorescent derivatives, 2D gel electrophoresis and mass spectrometry). In our manuscript, we show in Figure 2C that PACMA-31 is interacting with free thiols in PDI and other thiol isomerases, presumably through the formation of covalent bonds. Additionally, the results depicted in Figure 2D show independently that the inhibitory effect of PACMA-31 on platelet aggregation is irreversible. A possible explanation for the partial reversibility shown in Figure S3 (now S5) might be that insufficient PACMA-31 was present during the incubation step to fully block all free thiols. This would be reflected as a partial restoration of PDI activity after the dilution step, as was observed.

8. *Page 7, please provide references in the text to the suppl. Videos.*

We apologize for this oversight. The requested references have been added. (Page 8, first paragraph)

9. *Page 7 and Fig 3: the term "vehicle" is not provided in the methods. Only control before adding bepristats is mentioned. Please clarify this inconsistency.*

We agree that that this inconsistency can be confusing. In order to maintain consistency and maximize clarity, we have revised the manuscript to utilize the phrase "vehicle control" at all points when we are referring to the control used in the animal experiments.

10. *Page 7, correct "by 14.9%" to "to 14.9%".*

We thank the reviewer for this comment and altered the manuscript accordingly.

11. *Figure 5: In the B panel, please provide the sizes of the proteins in the gel and explain what exactly shown there in this picture in the text (page 9). In D panel, one can see only the red line; the blue line should be shown as well even though these lines merge. In the F panel, explain more clearly in the legend what is shown, at what ratio of the GSH/GSSG this fraction is measured? Is it a bar representation of one of the points shown in panel E?.*

Figure 5B and the text describing the proteinase K digestion have been adapted to provide more clarity (Page 9, line 6 and 7 last paragraph). In order to address the comments about the overlapping tracings in Figure 5D, two distinct figures depicting 'bepriostat 1' (left) and 'bepriostat 2' (right) were created to get rid of this potential problem. The legend of 'Figure 5F' has also been revised in accordance with the reviewer's comments.

12. *Page 11, the last line should be written more accurately. Only Cathepsin inhibited the catalytic activity of ERp5 and Erp57 according to Fig 6.*

We thank the reviewer for this correction, and the manuscript has been revised accordingly.

13. *Suppl. Fig S1: the Y title should be corrected to "PDI activity in %"*

Agreed. We have revised the manuscript accordingly.

14. *Suppl. Fig S2: what are the numbers (BRD1035, BRD4832)? It was not mentioned anywhere before - it should be consistent. The statistical significance should be shown in the graphs.*

We inadvertently retained a previous nomenclature. We have revised the legend accordingly and appreciate the comment.

15. *Suppl. Fig S4: this figure has no reference and explanation in the text. It can be either removed or be explained in the results and figure legend.*

We thank the reviewer for noting this oversight. An appropriate reference to Supplemental Fig. S4 (now S6) has been made in the last paragraph of page 7.

16. *Suppl. Fig S5: In the graph instead of "ANS" it should be "vehicle"*

We appreciate the care the reviewer has taken and have revised the manuscript accordingly.

Discussion

1. *In line 10 of the second paragraph the term "reductase activity" should be more specific. It was only inhibited in the insulin assay for these enzymes and not in the di-e-GSSG assay. The authors need to more clearly discriminate between these assays throughout the whole manuscript because these assays display opposite results and this can be confusing to the reader.*

We thank the reviewer for this well-taken point. We have revised the manuscript to better distinguish between the two assays and make our language more clear and consistent throughout.

2. *Line 11 of the same paragraph is not clear and need to be clarified.*

We thank the reviewer for this point and have revised the manuscript to improve clarity.

3. *Page 13, line 1 of the second paragraph: the word "also" appears twice. Erase the first one.*

Agreed. We have revised the manuscript accordingly.

4. *Page 13: The authors should address the finding that bepristats inhibited only PDI,*

while the other peptides also inhibit ERp57 and provide possible explanation for that.

Bacitracin is a cyclic peptide that adopts an amphipathic configuration upon binding substrates (PNAS, 110:14207). It interacts with membranes and other hydrophobic surfaces non-specifically and with relatively low affinity. It appears to interact with the hydrophobic surface of the substrate binding domains of thiol isomerases. This interaction has been shown with PDI. In addition to hydrophobic interactions, bacitracin forms a disulfide bond with free cysteines in the substrate-binding domain of PDI via its thiazoline ring (FEBS J, 278:2034). Formal possibilities for the ability of bacitracin to impair MPB binding to other thiol isomerases include (1) that at 150 μ M bacitracin has multiple binding sites and (2) bacitracin itself reacts with MPB. The fact that bacitracin inhibits MPB binding to ERp5 and ERp57, but is less effective against thioredoxin, favors the possibility that bacitracin has multiple binding sites as does the fact that none of the inhibitors interfered with MPB binding to BSA. PAMCA-31 is thiol reactive small molecule. Although it shows relative selectivity for PDI, it appears that at higher concentrations it reacts with the a domain and/or a' domain of other thiol isomerases. The b' domain demonstrates a higher degree of sequence variability among thiol isomerases than the a domain or a' domain. It is possible that critical amino acid side chains within the hydrophobic domain that required for bepristat binding are missing from other thiol isomerases. Future studies will be required to evaluate the nature of this specificity in detail. Although the activity of cathepsin G against ERp57 was modest (not reaching 50% inhibition at 1 μ M), it inhibited ERp5 relatively efficiently (Fig. 6c). One possibility is that cathepsin G, a serine protease could cleave ERp5, which contains several potential cathepsin G cleavage sites.

5. Page 13, last line: the fig should be 6d instead of 5d.

We thank the reviewer for this comment and revised the manuscript accordingly. (Page 14, second paragraph, line 2)

G. References are appropriate.

H. *Abstract and conclusions are appropriate, except for some issues that should be addressed in the "Discussion" as detailed in point F. The introduction needs some improvement. It should contain more details about the other thiol isomerases mentioned in the results and what is known about other peptides that bind the b' domain like mastoparan. These peptides are firstly mentioned in the results section even though they present the same binding pattern as the bepristats. In general, the authors should go through the results section and provide some of the previously known data that is first mentioned there in the introduction section instead.*

We now include in the Introduction the other thiol isomerases and peptides used in this manuscript (Page 4, last two sentences of first paragraph). To maintain focus on the overarching concepts of the manuscript, we did not include a detailed background of these reagents. However, we do provide references in the introduction that provide this background information for the interested reader. We have modified the Discussion as addressed in the response to comments in section F.

Reviewer #3:

Suggestion: the authors should do a better job of explaining the how the observed and hypothesized structural effects of the BPSs can be rationalized with both the inhibition thrombus and insulin turbidity and paradoxically the activation of reductase activation. Perhaps they can make use of this reviewers comments above.

We thank the reviewer for this important comment. The manuscript has been revised to

provide a more robust explanation for this paradox. (Page 13, last four sentences, last paragraph)

Minor point: Fig 2a-black line is missing in the legend for PDI.

We thank the reviewer for this comment and have revised the manuscript accordingly. (Figure 2a has been updated)

We feel that the reviewers' comments have meaningfully improved the manuscript and hope that it is now acceptable for publication in *Nature Communications*.

Sincerely,

Robert Flaumenhaft

Reviewer #1 (Remarks to the Author): After revision a significant paper.

Reviewer #3 (Remarks to the Author): The authors' revisions are satisfactory. As a result the MS is much improved.

Editorial Note

Reviewer #2 communicated to the editor that the comments raised in the report have been satisfactorily addressed by the authors and that the revised paper is appropriate for publication.